



# Aerosol acidity and liquid water content regulate the dry deposition of inorganic reactive nitrogen

Athanasios Nenes[1,2*], Spyros N. Pandis[1,3], Maria Kanakidou[1,4], Armistead Russell[5], Shaojie Song[6], Petros Vasilakos[5], Rodney J. Weber[7]

[1]Institute for Chemical Engineering Sciences, Foundation for Research and Technology Hellas, Patras, GR-26504, Greece

[2]School of Architecture, Civil & Environmental Engineering, Ecole Polytechnique Fédérale de Lausanne, CH-1015, Lausanne, Switzerland

[3]Department of Chemical Engineering, University of Patras, GR-26504, Greece

[4]Environmental Chemical Processes Laboratory, Department of Chemistry, University of Crete, Voutes, Heraklion Crete, 70013, Greece

[5]School of Civil & Environmental Engineering, Georgia Institute of Technology, Atlanta, GA 30332, USA

[6] School of Engineering and Applied Sciences, Harvard University, Cambridge, Massachusetts
02138, USA

[7]School of Earth and Atmospheric Sciences, Georgia Institute of Technology, Atlanta, GA 30332, USA

*correspondence to athanasios.nenes@epfl.ch*

**Abstract.** Ecosystem productivity is strongly modulated by the atmospheric deposition of inorganic reactive nitrogen (the sum of ammonium and nitrate). The individual contributions of ammonium and nitrate vary considerably over space and time, giving rise to complex patterns of nitrogen deposition. In the absence of rain, much of this complexity is driven by the large
difference between the dry deposition velocity of nitrogen-containing molecules in the gas or condensed phase. Here we quantify how aerosol liquid water and acidity, through their impact on gas-to-particle partitioning, modulate the deposition velocity of $NH_3$ and $HNO_3$ individually, while simultaneously affecting the dry deposition of inorganic reactive nitrogen. Four regimes of deposition velocity emerge: *i*) $HNO_3$-fast, $NH_3$-slow, *ii*) $HNO_3$-slow, $NH_3$-fast, *iii*) $HNO_3$-fast,
$NH_3$-fast, and, *iv*) $HNO_3$-slow, $NH_3$-slow, Conditions that favor partitioning of species to the aerosol phase strongly reduce the deposition of reactive nitrogen species and promote their accumulation in the boundary layer and potential for long-range transport. Application of this framework to select locations around the world reveals fundamentally important insights: The dry deposition of total ammonia displays little sensitivity to pH and liquid water variations, except





under conditions of extreme acidity and/or low aerosol liquid water content. The dry deposition of total nitric acid, on the other hand, is quite variable, with maximum deposition velocities (close to gas-deposition rates) found in the Eastern US and minimum velocities in Northern Europe and China. In the latter case, the low deposition velocity leads to up to 10-fold increases in $PM_{2.5}$ nitrate

5 aerosol, thus contributing to the high $PM_{2.5}$ levels observed during haze episodes. In this light, aerosol pH and associated liquid water content can be considered as control parameters that drive dry deposition flux and can accelerate the accumulation of aerosol contributing to intense haze events throughout the globe.



# 1. Introduction

Human civilization and a high standard of living fundamentally depend on a sustainable food and energy supply (NRC, 2016). Both can be linked to the availability of reactive nitrogen ($N_r$) species in agricultural soils, which are key nutrients required by the biosphere for growth and crop production. Although fertilizers rich in $N_r$ have ensured food security for most of the global population, riverine outflow and atmospheric deposition of $N_r$ to oceans and coastal zones has increased roughly three-fold compared to preindustrial levels (Duce et al., 2010; Kanakidou et al., 2018), with profound and diverse impacts on the Earth System (Ito et al., 2016; Jickells et al., 2017; Suntharalingam et al., 2019). $N_r$ includes reduced nitrogen species – the main components of which are gas-phase ammonia ($NH_3$) and its particulate-phase counterpart ammonium ($NH_4^+$). Oxidized nitrogen also constitutes a significant fraction of $N_r$, being primarily in the form of nitric acid ($HNO_3$) and particulate-phase nitrate ($NO_3^-$) (Seinfeld and Pandis, 2016).

Reactive inorganic nitrogen species are major components of ambient particulate matter (Kanakidou et al., 2005; Sardar et al., 2005; Zhang et al., 2007), with important implications for human health (Pope et al., 2004; Lim et al., 2012; Lelieveld et al., 2015; Cohen et al., 2017), ecosystem productivity (Fowler et al., 2013) and the climate system (Haywood and Boucher, 2000; Bellouin et al., 2011; IPCC, 2013). Ammonia reacts with sulfuric and nitric acids to form ammonium sulfate/bisulfate and nitrate aerosol, while gas-phase nitric acid can also react with non-volatile cations found in sea salt, mineral dust and biomass burning to form a variety of inorganic (soluble) salts. The acidity (pH) and liquid water content of aerosol in turn are central parameters that govern the gas-particle partitioning of $N_r$ species (Meskhidze et al., 2003; Guo et al., 2015; Guo et al., 2018; Nenes et al., 2020). Given that species in the gas phase generally have a different atmospheric residence time than those in the aerosol phase (Seinfeld and Pandis, 2016), the degree of gas-particle partitioning can directly impact the atmospheric residence time of $N_r$ species, with important implications for particulate matter levels and dry deposition (e.g., Vayenas et al., JGR, 2005; Pinder et al., 2007, 2008). It is therefore important to consider how aerosol acidity and liquid water content, through their effect on gas-to-particle partitioning, can affect the deposition fluxes of $N_r$ species. Here we present a simple and thermodynamically consistent framework to address the links between deposition of $N_r$ and aerosol acidity. We then demonstrate the power of this new framework with observational data to understand the relevant "chemical





regimes" that apply to select places around the world, and to understand the implications that changes in acidity have on deposition flux and PM levels.

## 2. The new conceptual framework

Nenes et al. (2020), based on the work of Meskhidze et al. (2003), Guo et al. (2016) and
Guo et al. (2017) developed a framework, where the sensitivity of PM to emissions of ammonia and nitrate is expressed in terms of aerosol pH and liquid water content. The basis of this approach lies in the realization that the magnitude of PM sensitivity to precursor emissions is driven by gas-to-particle partitioning, the latter of which is controlled by aerosol acidity and liquid water content. Given that the deposition velocity of ammonia and nitrate also is strongly modulated by the same
partitioning, it should show similar sensitivity to the pH and liquid water content.

### 2.1 Linking deposition flux with partitioning fraction

For a given air mass with total nitrate $NO_3{}^T$ (i.e., the amount of aerosol and gas-phase nitrate), the equilibrium aerosol nitrate concentration, $c_{NO_3^-}$, is given by $c_{NO_3^-} = \varepsilon(NO_3^-)\,NO_3^T$, where $\varepsilon(NO_3^-)$ is the fraction of $NO_3{}^T$ that partitions to the particle phase. The deposition flux of
$NO_3{}^T$, $F_{NO_3^T}$, is then given by the contribution from the gas and particle phases (Seinfeld and Pandis, 2016):

$$F_{NO_3^T} = v_g c_{HNO_3} + v_p c_{NO_3^-} = v_p\{k + (1-k)\varepsilon(NO_3^-)\}NO_3^T \tag{1}$$

where $F_{NO_3^T}$ has units of moles N per unit area and time, $v_g$, $v_p$ are the gas- and particulate-phase deposition velocities, respectively and $k = v_g/v_p$ (of order 10 for $NH_3^T$ and $NO_3{}^T$).

Similarly, equilibrium partitioning of $NH_3^T$ to the aerosol is given by $NH_4^+ = \varepsilon(NH_4^+)\,NH_3^T$,
where $\varepsilon(NH_4^+)$ is the fraction of $NH_3^T$ (i.e., the amount of aerosol ammonium and gas-phase ammonia) that partitions to the particle phase:

$$F_{NH_3^T} = v_p\{k + (1-k)\varepsilon(NH_4^+)\}NH_3^T \tag{2}$$

where $F_{NH_3^T}$ has units of moles N per unit area and time and $\varepsilon(NH_4^+)$ is the fraction of $NH_3^T$ (i.e., the amount of aerosol ammonium and gas-phase ammonia) that partitions to the particle phase.





### 2.2 The acidity and liquid water content link: deposition velocity "regimes"

From equations (1) and (2), it becomes apparent that the $\varepsilon(NH_4^+)$ and $\varepsilon(NO_3^-)$ are modulators of the deposition fluxes. Depending on their value, both $F_{NH_3^T}$ and $F_{NO_3^T}$ vary by a factor of 10 with the highest values corresponding to partitioning fractions close to zero. Through

their effects on gas-particle partitioning, aerosol acidity and liquid water content can impact the deposition flux, which we quantify as follows.

The partitioning fractions can be expressed as functions of the liquid water content, $W_i$:

$$\varepsilon(NO_3^-) = \frac{\Psi W_i}{[H^+] + \Psi W_i} \qquad\qquad \varepsilon(NH_4^+) = \frac{\Phi[H^+]W_i}{1 + \Phi[H^+]W_i} \qquad (3)$$

where $\Psi = \frac{RTK_{n1}H_{HNO_3}}{\gamma_{H^+}\gamma_{NO_3^-}}$ and $\Phi = \frac{\gamma_{H^+}}{\gamma_{NH_4^+}}\frac{H_{NH_3}}{K_a}RT$ following the definitions of Nenes et al. (2020).

Equations (3) yields "sigmoidal" functions, with the partitioning fraction ranging from zero to

unity over a characteristic pH range. The same equation can then be used to define a "characteristic pH" that determines when the deposition of ammonia or nitrate is "fast" (i.e., effective deposition velocity approximately equal to that of the gas phase) or "slow" (i.e., approximately equal to the particle-phase deposition velocity). Following Nenes et al. (2020), we define characteristic thresholds being $\alpha$ for $\varepsilon(NO_3^-)$ and $\beta$ for $\varepsilon(NH_4^+)$, which separate the corresponding velocity

deposition regimes (Figure 1). When $\varepsilon(NO_3^-)$ is below $\alpha$ (or $\varepsilon(NH_4^+)$ is below $\beta$), a sufficient amount of $NO_3^T$ (or $NH_3^T$) is in the gas phase so that the effective deposition velocity is approximately equal to the gas-phase limit ("fast" for $NO_3^T$ or $NH_3^T$, respectively). When the partitioning fractions are above their respective threshold, the deposition velocity approaches the aerosol limit ("slow"). Based on these considerations $k\sim10$ and $\alpha=\beta=0.9$ are used to define the

thresholds. Equations (5) and (6) of Nenes et al. (2020) then give a "characteristic" acidity level $pH' = -\log[0.11\Psi W_i]$ for nitrate and $pH'' = \log[0.11\Phi W_i]$ for ammonium. Figure 2 displays $pH'$ and $pH''$ for 273 K and 298 K; $pH'$ tends to decrease with increasing $W_i$, and vice-versa for ammonium and $pH''$.

Based on the values of $pH'$, $pH''$ and its relation to the aerosol pH, we can then determine

whether the deposition $NO_3^T$ (or $NH_3^T$) is "slow" or "fast"; $N_r$ deposition can belong to one of four distinct chemical regimes:





- Regime 1: pH $> pH''$ and pH $< pH'$: "NH₃ fast, HNO₃ fast".
- Regime 2: pH $> pH''$ and pH $> pH'$: "NH₃ fast, HNO₃ slow".
- Regime 3: pH $< pH''$ and pH $< pH'$: "NH₃ slow, HNO₃ slow".
- Regime 4: pH $< pH''$ and pH $< pH'$: "NH₃ slow, HNO₃ fast".

Figure 2 indicates these four regions. Observation or modeling data, based on their value of acidity and liquid water, will belong to one of these four "chemical regimes" of deposition velocity for $NH_3^T$ and/or $NO_3^T$.

Following Nenes et al. (2020), the characteristic point (defined by a characteristic acidity $pH^*$ and liquid water content $W_i^*$) on the chemical regime map where the two lines "crossover",

thus separating Regime 1 from Regime 3, and Regime 2 from Regime 4 corresponds to a $pH^* = -\frac{1}{2} log \left( \frac{\Psi}{\Phi} \right) \sim 2.2$, and $W_i^* \sim 300$ µg m⁻³ for 298K. Therefore, for moderately acidic aerosol (pH*~2) and sufficiently high amounts of water – the aerosol levels can considerably increase because the deposition velocity changes by about 10-fold for both $NH_3^T$ and $NO_3^T$. For higher (or lower) pH levels, the aerosol transitions between regions 2 (or 4). For liquid water above $W_i^*$, there is a

"transition pH" from a NH₃-slow, HNO₃-fast deposition scheme to a NH₃-fast, HNO₃-slow deposition rates (Fig. 2). Similarly, there is also another "transition pH" that defines when both $NH_3^T$ and $NO_3^T$ change from "fast" to "slow" deposition. Given the combined complexity of deposition velocity and aerosol thermodynamics, it is remarkable that deposition velocity patterns for $NH_3^T$ and $NO_3^T$ can be unraveled simply using pH and liquid water. This is illustrated in the

following section.

### 3. Application of the framework

The above framework requires knowledge of aerosol pH and liquid water content, which can be routinely simulated by state-of-the-art atmospheric chemical transport models (e.g., CMAQ, CAMx). Thermodynamic analysis of ambient aerosol and gas-phase measurements also provides

aerosol pH and liquid water content. Therefore, the above framework can be used to characterize the chemical domain of both ambient and simulated aerosol.

Application of the chemical domain approach is demonstrated for ambient data representing select locations over the world. For this purpose, we use the same datasets as in Nenes et al. (2020), which were obtained from observations over 5 locations worldwide: Cabauw,





Netherlands (CBW), Tianjin, China (TJN), California, USA (CNX), SE US (SAS), and a wintertime NE USA (WIN) study. This dataset covers a wide range of atmospheric acidity, temperature, relative humidity and liquid water levels. Figure 3 presents the chemical domain classifications for each location, with characteristic curves being calculated from the average
temperature of the dataset. Cabauw is characterized by high deposition velocity for $NH_3^T$, while $NO_3^T$ alternates between fast and slow deposition, with most of the time being in the slow deposition regime (Figure 3a). The California dataset (Figure 3b) is quite interesting, as both $NH_3^T$ and $NO_3^T$ deposit rapidly, which means that they are less subject to long-range transport and are rapidly lost from the boundary layer. Tianjin displays similar behavior to Cabauw (Figure 3c). The
Southeast US (SAS) is considerably more acidic; for this reason $NH_3^T$ exhibits variability in its deposition velocity (alternating between slow and fast), while $NO_3^T$ always maintains a high deposition velocity. The wintertime eastern US dataset (WIN) corresponds to a broad region (aircraft data set), hence the data naturally occupies multiple domains. There are no locations in this dataset characterized by low deposition velocities for both $NH_3^T$ and $NO_3^T$.

**4. Implications for deposition flux of $N_r$ at selection locations worldwide**

To understand the implications of acidity and liquid variations on the reactive nitrogen flux (moles m$^{-2}$ s$^{-1}$), we first sum the contributions from $NO_3{}^T$ and $NH_3{}^T$. Summing Equations 1 and 2 yields the reactive nitrogen flux, $F_{N_r}$ , expressed as mols N per area and unit time:

$$F_{N_r} = v_p\{k + (1 - k)\varepsilon(NH_4^+)\}NH_3^T + v_p\{k + (1 - k)\varepsilon(NO_3^-)\}NO_3^T \tag{4}$$

Defining $\Gamma = \frac{NH_3^T}{NH_3^T + NO_3^T}$ (fraction of N$r$ that is $NH_3^T$) and substituting into Equation 4 gives:

$$F_{N_r} = v[(1 - k)\{\varepsilon(NH_4^+) - \varepsilon(NO_3^-)\}\Gamma + \{k + (1 - k)\varepsilon(NO_3^-)\}]N_r \tag{5}$$

Further dividing Equation (5) with $v_p N_r$ gives the non-dimensional N$_r$ flux, $F_{N_r}^*$ :

$$F_{N_r}^* = \frac{F_{N_r}}{v_p N_r} = (1 - k)\{\varepsilon(NH_4^+) - \varepsilon(NO_3^-)\}\Gamma + \{k + (1 - k)\varepsilon(NO_3^-)\} \tag{6}$$


$F_{N_r}^*$ expresses how rapid $F_{N_r}$ is, compared to if the flux of $N_r$ occurred with the particle deposition

velocity. Equation 6 embodies the effect of acidity and liquid water content (through their effect

on $\varepsilon(NH_4^+)$, and $\varepsilon(NO_3^-)$) on the $N_r$ deposition velocity and flux. $F_{N_r}^*$ ranges between 1 and $k$;

when it is equal to $k$, deposition is most efficient and occurs with the gas-phase deposition velocity

for both $NO_3^T$ and $NH_3^T$ (Regime 1). Under such conditions, reactive nitrogen is rapidly lost from

the atmosphere. When $F_{N_r}^*$ is unity, all the reactive nitrogen is in particulate form and the dry

deposition is at its slowest possible rate (Regime 4) – allowing its transport over larger distances

and resulting in an increase of its lifetime in the boundary layer (considering only dry removal) by

a factor of $k$.

10       We apply Equation 6 on observations that span a wide range of atmospheric acidity and

liquid water content, to explore the degree to which atmospheric acidity can modulate $F_{N_r}^*$. The

observations analyzed (Table 1) are for characteristic conditions found in the US, Europe and

China (refer to Guo et al. (2017) for a more thorough analysis). US and European sites cited tend

to be more acidic than the Asian sites (although Cabauw, clearly is similar to the latter, Figure 3a).

To understand the deposition rate trends, we first examine the patterns for ammonia and nitrate

separately, and then the combined $N_r$ flux.

For ammonia, given that the pH is mostly above 1 (Table 1), the corresponding partitioning

fractions are below 0.9 (Figure 4a) and its deposition is generally "fast". Indeed, its non-

dimensional deposition flux (Figure 4b) is often above 5 and in polluted conditions in China it

reaches values as high as 9 (Figure 4b); lower values are only seen for the most acidic and warm

conditions for the SE US. The latter means that ammonia has a larger lifetime in the boundary

layer of the SE US, allowing for its buildup and transport over larger distances, relatively to the

situation in China, North Europe and other locations with mildly acidic aerosol.

Nitric acid shows a different behavior, effectively exhibiting the full range of partitioning fractions

and deposition velocities for the sites studied (Figures 4ab), giving a normalized deposition flux

of ~10 (i.e., mainly deposited as gaseous $HNO_3$) in the SE US and Greece, and ~1 for the Asian

sites. The immediate implication for its deposition patterns is that at the more acidic sites, $NO_3^T$

remains close to the production region, while for less acidic conditions it can be transported away

with a 10-fold increase in its boundary layer lifetime.



To understand the impact of both $NH_3$ and $HNO_3$ pattern variability on the deposition flux of total reactive nitrogen, we first note that $NH_3^T$ constitutes the majority of the $N_r$ in all sites considered (i.e., $\Gamma \sim 0.5$-$0.85$; Table 1). Given this, and taking into account that the acidity levels usually do not approach low enough levels to ensure that $\varepsilon(NH_4^+) \rightarrow 1$ (Figure 3a), $F_{N_r}$ does not

substantially vary from location to location, and, it closely follows that of $NH_3$ (Figure 4b). Because of the opposite trends of normalized deposition flux between $NH_3^T$ and $NO_3^T$ (Figure 4b), the shifts in ammonia deposition flux are partially mitigated when included in $F_{N_r}^*$. The implication is that, although nitrate exhibits large variability in dry deposition flux rate, ammonia should exhibit less variability – and therefore total reactive nitrogen should also have lower

variability.

## 5. Implications of modified deposition flux for boundary-layer PM$_{2.5}$ levels

The immediate consequence of modulating deposition velocities is that the atmospheric residence time of aerosol precursors is changed. When the deposition velocity of $N_r$ species is low,

then the precursor is given an opportunity to accumulate – when it also corresponds to a compound that PM$_{2.5}$ levels are sensitive to, aerosol levels can increase considerably. Based on the discussion in Section 4, aerosol nitrate accumulates (i.e., "slow deposition"), exactly when aerosol is most sensitive to changes in its concentration. To illustrate this effect, we estimate how the concentration of aerosol nitrate and ammonium would change in response to changes in deposition

velocity. Following the approach of Weber et al. (2016), the steady-state boundary layer concentration of nitrate can be written as:

$$NO_3^T = \frac{h}{v_p} \frac{E + P}{\{k + (1 - k)\varepsilon(NO_3^-)\}} \qquad (7)$$

where $h$ is the height of the boundary layer, $E$ is the emission rate including transport from outside the boundary layer, and $P$ is the photochemical production rate. Assuming that partitioning of total nitrate changes from a value $\varepsilon(NO_3^-)_{(1)}$ to $\varepsilon(NO_3^-)_{(2)}$, the steady-state $NO_3^T$ (with all other factors

being equal) changes according to:



$$\mathrm{NO}_{3,(2)}^{\mathrm{T}} = \frac{\{k + (1-k)\varepsilon(\mathrm{NO}_3^-)_{(1)}\}}{\{k + (1-k)\varepsilon(\mathrm{NO}_3^-)_{(2)}\}} \mathrm{NO}_{3,(1)}^{\mathrm{T}} \qquad (8)$$

Equation 8 suggests that if the pH (through its impact on $\varepsilon(\mathrm{NO}_3^-)$) varies sufficiently, the change in total nitrate in the boundary layer will approach a factor of $k{\sim}10$ (Figure 4a). Assuming that this occurs, we can then show PM would drop considerably in regions where nitrate constitutes a significant fraction of PM levels (Figure 4a, difference between solid lines and dotted lines).

Given that aerosol nitrate is often a significant constituent of PM in regions where the aerosol exhibits a mildly acidic pH (e.g., Cabauw, China), this constitutes an important and overlooked positive feedback between acidity, nitric acid production, ammonium and PM buildup that is a consequence of aerosol thermodynamic partitioning and deposition velocity. These processes are summarized in Figure 5. A time varying mixing layer, along with storage of both $\mathrm{NH}_3^{\mathrm{T}}$ and

$\mathrm{NO}_3^{\mathrm{T}}$ aloft, complicate the dynamics, but the implications are the same: pH and liquid water content are integral to the deposition of reactive nitrogen.

## 6. Conclusions

Here we present a simple framework to understand how aerosol acidity and liquid water

content can modulate the dry deposition flux of inorganic reactive nitrogen species, and assessment of factors that cause variability in $N_r$ deposition and the implications. Our analysis identifies four deposition velocity regimes: *i)* HNO₃-fast, NH₃-slow, *ii)* HNO₃-slow, NH₃-fast, *iii)* HNO₃-fast, NH₃-fast, and, *iv)* HNO₃-slow, NH₃-slow. When this framework is applied to ambient measurements or predictions of PM and gaseous precursors, the "chemical regime" of deposition

velocity is directly determined. Generally, conditions that favor strong partitioning of species to the aerosol phase strongly impact the deposition flux of $N_r$ species and their potential for long-range transport. Applying this framework to select locations around the world reveals important implications: *i)* ammonia deposition rates display little sensitivity to pH and liquid water variations, except under conditions of extreme acidity or aerosol liquid water content, *ii)* for the examples

considered dry deposition of total $N_r$ is 50-85% (by mol) total ammonia, which means that modest modulations of $N_r$ flux are seen from acidity changes. There are regions, however, (e.g., coastal areas, dust) where total nitrate may constitute a much larger fraction of the $N_r$ – and for which acidity would drive fluctuations thereof. Total nitrate deposition flux is close to the maximum



levels (approximately equal to the corresponding gas-deposition rates) in the Eastern US and has minimum rates for North Europe and China. When the latter occurs, the low deposition velocity can promote considerable accumulation of nitrate aerosol in the boundary layer – increasing up to 10-fold the PM$_{2.5}$ nitrate, eventually causing the extremely high levels observed. If liquid water

content is high enough (e.g., close to fog conditions) both total ammonia and nitrate deposit slowly and are allowed to accumulate, which create the potential for the most intense haze episodes.

       The above also point to the effect that model prediction biases in pH could have when it comes to prediction of reactive nitrogen deposition fluxes and PM levels in the boundary layer. For example, in the case of PM$_{2.5}$ nitrate, not only the aerosol levels will be directly affected owing

to partitioning ratio biases (Vasilakos et al., 2018) but too much accumulation (if pH is predicted too high) or accelerated loss (if pH is predicted too low) may magnify nitrate prediction biases through the discussed feedback from deposition velocity. A similar situation can also arise for aerosol ammonium, only that the trends of bias with respect to pH are reversed. Identical biases also can arise for semi-volatile species that are sensitive to aerosol pH, such as chloride, amines

and organic acids.

       Although ammonium almost exclusively resides in the fine mode – where the equilibrium analysis carried out here is most applicable - nitrate can also reside partially in the coarse mode in the presence of dust and/or sea salt particles (e.g., Karydis et al., 2016) because the latter tend to have a high enough pH to favor partitioning (Vasilakos et al., 2018; Fang et al., 2017). The same

principle and trends also apply in such situations, as acidity and liquid water content drive gas-to-aerosol partitioning and modulate the deposition velocity (the analysis can also be extended to separately consider coarse and fine mode particles), though larger particles have a higher deposition velocity than submicron particles. The effect of organic aerosol and its associated liquid water content may also affect the partitioning, although such effects are most likely limited to low

relative humidity (below 40%) and temperatures low enough for strong diffusivity limitations to limit the applicability of equilibrium considerations (Battaglia Jr. et al., 2019; Pye et al., 2020). Despite these limitations, the new understanding presented here applies throughout most of the atmosphere close to the surface demonstrating that aerosol pH and its associated liquid water content naturally emerge as parameters that drive dry deposition flux and atmospheric lifetime of

reactive nitrogen, and can initiate feedbacks that promote heavy PM$_{2.5}$ pollution episodes.





**Acknowledgements**

This work was supported by the project PyroTRACH (ERC-2016-COG) funded from H2020-EU.1.1. - Excellent Science - European Research Council (ERC), project ID 726165. MK acknowledges support of this work by the project "PANhellenic infrastructure for Atmospheric Composition and climatE change" (MIS 5021516), which is implemented under the Action "Reinforcement of the Research and Innovation Infrastructure", funded by the Operational Programme "Competitiveness, Entrepreneurship and Innovation" (NSRF 2014–2020) and co-financed by Greece and the European Union (European Regional Development Fund). RW acknowledges support from the U.S. EPA under grant R83588201. AGR acknowledges support from the U.S. EPA under grants R83588001 and R83588201. Its contents are solely the responsibility of the grantee and do not necessarily represent the official views of the supporting agencies. Further, the U.S. government does not endorse the purchase of any commercial products or services mentioned in the publication.

**Code and Data availability**

User access to data used in this manuscript is described in the citations referenced for each dataset, and can also be accessed from the compiled dataset of Pye et al. (2019). The ISORROPIA-II thermodynamic equilibrium code is available at isorropia.epfl.ch. Relevant spreadsheets and mapping templates are provided upon request.

**Competing interests**

The authors declare that they have no conflicts of interest.

25 **Author contributions**

AN initiated the study, developed the framework, carried out analysis of the ambient data and wrote the initial draft. All authors helped interpret the data, provided feedback on the analysis approach and extensively commented on the manuscript.

30



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





**Table 1.** Aerosol composition and meteorological conditions for flux deposition calculations, and aerosol composition changes from modifications to the deposition velocity. The reported concentrations and RH, T are obtained from Guo et al. (2017), table S1. The data in rows 6-15 are input to ISORROPIA-II (metastable, forward) and the output is then presented in rows 16-20. This data and the equations presented in the text is utilized to compute the quantities remaining columns (21-28).

| Region/Location | SE US | SW US | | Greece | Beijing, China | | Xi'an, China | |
|---|---|---|---|---|---|---|---|---|
| Sampling type | Ground | Ground | Ground | Ground | Ground | | Ground | |
| PM cut size | $PM_1$ & $PM_{2.5}$[a] | $PM_1$ | $PM_{2.5}$[c] | $PM_1$ | $PM_1$ | | $PM_{2.5}$ | |
| Year | 2013 | 2010 | | 2012 & 2014 | 2013 | | 2013 | |
| Season | Summer | (Early) Summer | | Summer &Winter | Winter | | Winter | |
| $Na^+$, µg m$^{-3}$ | 0.03 | 0 | 0.77 | 0.08 | 0 | 0 | 3.6 | 4.2 |
| $SO_4^{2-}$, µg m$^{-3}$ | 1.73 | 2.86 | 1.88 | 1.66 | 4.2 | 14 | 5.9 | 38 |
| Total $NH_4^+$, µg m$^{-3}$ | 0.78 | 3.44 | 2.54 | 1.02 | 9.5 | 33.5 | 13 | 44.3 |
| Total $NO_3^-$, µg m$^{-3}$ | 0.45 | 10.22 | 8.19 | 3.36 | 6.6 | 18 | 8.7 | 33 |
| $Cl^-$, µg m$^{-3}$ | 0.02 | 0 | 0.64 | 0.20 | 0.8 | 1.6 | 4.0 | 14 |
| $Ca^{2+}$, µg m$^{-3}$ | 0 | 0 | 0 | 0 | 0 | 0 | 1.6 | 2.3 |
| $K^+$, µg m$^{-3}$ | 0 | 0 | 0 | 0.36 | 0 | 0 | 1.3 | 4.6 |
| $Mg^{2+}$, µg m$^{-3}$ | 0 | 0 | 0 | 0 | 0 | 0 | 0.2 | 0.3 |
| RH, % | 74 | 79 | 87 | 68 | 40e | 56 | 46 | 68 |
| T, °C | 25 | 18 | 18 | 20 | 0.4 | 0.9 | 5.7 | 4.1 |
| $NH_3$, µg m$^{-3}$ | 0.21 | 1.06 | 0.86 | 0.62 | 5.92 | 23.04 | 11.54 | 23.17 |
| $NH_4^+$, µg m$^{-3}$ | 0.60 | 2.52 | 1.78 | 0.43 | 3.79 | 11.07 | 1.55 | 22.37 |
| $NO_3^-$, µg m$^{-3}$ | 0.00 | 5.11 | 5.37 | 0.16 | 6.46 | 17.71 | 8.56 | 32.47 |
| Calculated pH | 0.83 | 1.63 | 2.61 | 1.90 | 3.90 | 4.52 | 5.50 | 4.71 |
| LWC | 2.39 | 12.41 | 22.30 | 1.65 | 4.25 | 21.85 | 10.21 | 100.20 |
| $\varepsilon_{NO3}$ | 0.01 | 0.50 | 0.66 | 0.05 | 0.98 | 0.98 | 0.98 | 0.98 |
| $\varepsilon_{NH3}$ | 0.77 | 0.73 | 0.70 | 0.42 | 0.40 | 0.33 | 0.12 | 0.50 |
| $\Gamma$ | 0.86 | 0.55 | 0.53 | 0.53 | 0.84 | 0.87 | 0.84 | 0.83 |
| $F^*_T$ | 4.02 | 4.34 | 3.88 | 7.82 | 5.58 | 6.27 | 7.72 | 4.72 |
| $F^*_{NH3}$ | 3.09 | 3.40 | 3.69 | 6.23 | 6.41 | 7.03 | 8.93 | 5.46 |
| $F^*_{NO3}$ | 9.95 | 5.50 | 4.10 | 9.57 | 1.19 | 1.15 | 1.14 | 1.14 |
| $NO_3^-$, µg m$^{-3}$ (gas dep. rate) | 0.00 | 2.81 | 2.20 | 0.15 | 0.77 | 2.03 | 0.98 | 3.72 |
| $NH_4^+$, µg m$^{-3}$ (gas dep. rate) | 0.18 | 0.86 | 0.66 | 0.27 | 2.43 | 7.78 | 1.38 | 12.20 |



**Figure Captions**

**Figure 1.** Particle phase fraction of total nitrate, $\varepsilon(NO_3^-)$ (blue curve) and total ammonium, $\varepsilon(NH_4^+)$ (red curve) versus pH for a temperature of 288 K and an aerosol liquid water content of (a) 10 μg m$^{-3}$, and, (b) 250 μg m$^{-3}$. A combination of moderate acidity and low liquid water content creates conditions for rapid dry deposition of both total ammonium and nitrate (panel a) and vice versa for moderate acidity and high liquid water content (panel b). In defining the sensitivity domains, we have assumed that a partitioning fraction of 90% (black dotted lines), and its corresponding "characteristic" pH, defines where the aerosol deposition velocity dominates the total dry deposition rate of each nitrogen-containing species.

**Figure 2.** Chemical domains of deposition velocity for ammonia and nitrate. Shown are results for 273 K (panel a) and 298 K (panel b).

**Figure 3.** Chemical domains of deposition flux NH$_3$ and HNO$_3$ for 5 regions examined: a) Cabauw - CBW, b) CalNex - CNX, c) Tianjin – TJN, d) SOAS – SAS, and, e) E. United States (WIN).

**Figure 4.** Acidity/aerosol water impacts on aerosol concentration and deposition flux in select regions of the world that represent strongly acidic conditions (SE US), intermediate acidity conditions (SW US, Greece) and mildly acidic conditions (China). Shown is (a) the concentration of aerosol ammonium and nitrate, and the impact of deposition velocity determined by the aerosol acidity state, (b) impact of acidity state on normalized ammonia, nitrate and total reduced nitrogen flux.

**Figure 5.** Summary sketch of the interactions between aerosol pH and emissions of total ammonium and nitrate (a) aerosol pH is low and LWC is moderate, as is characteristic of the SE US. Here the partitioning of ammonium is mostly in the aerosol phase, and the relevant dry deposition velocity is low. The concentration of total ammonium is dictated by the aerosol deposition velocity limit; ammonia export to the free troposphere is favored, and vice versa for total nitrate. (b) aerosol pH is high and LWC is moderate (as is characteristic of N.Europe in the winter and China). Here the partitioning of total ammonia is shifted to the gas phase, and the relevant dry deposition velocity is rapid, total ammonia does not accumulate considerably in the boundary layer and export to the free troposphere is minimal. Nitrate partitions to the aerosol phase, desposits slowly and accumulates rapidly in the boundary layer. These conditions favor haze events and export to the free troposphere.





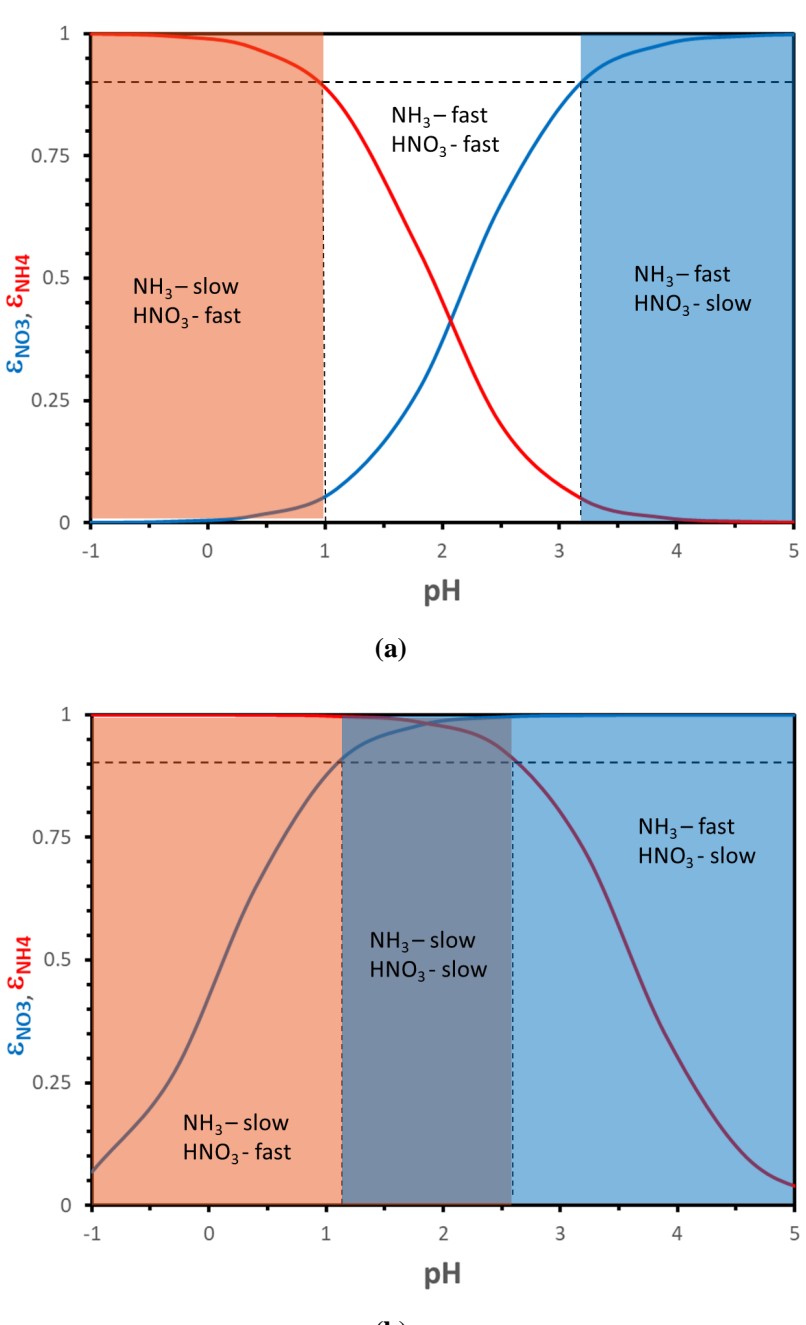

**Figure 1.**

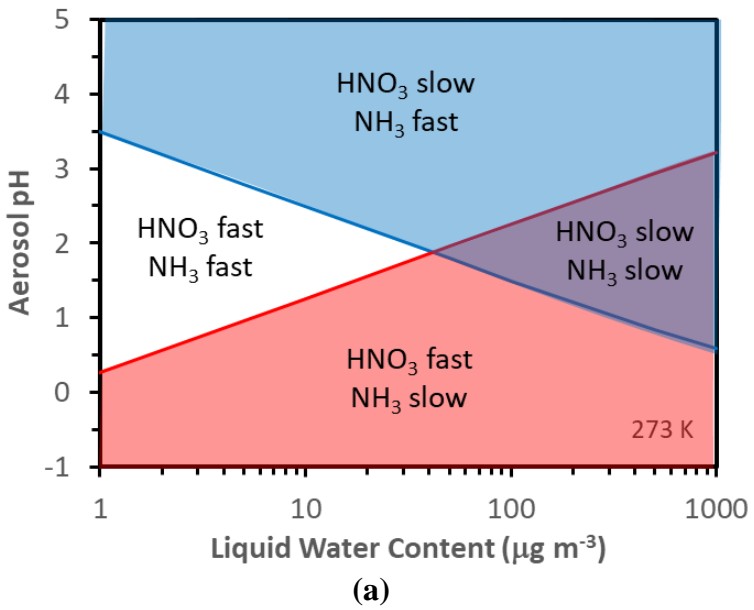

(a)

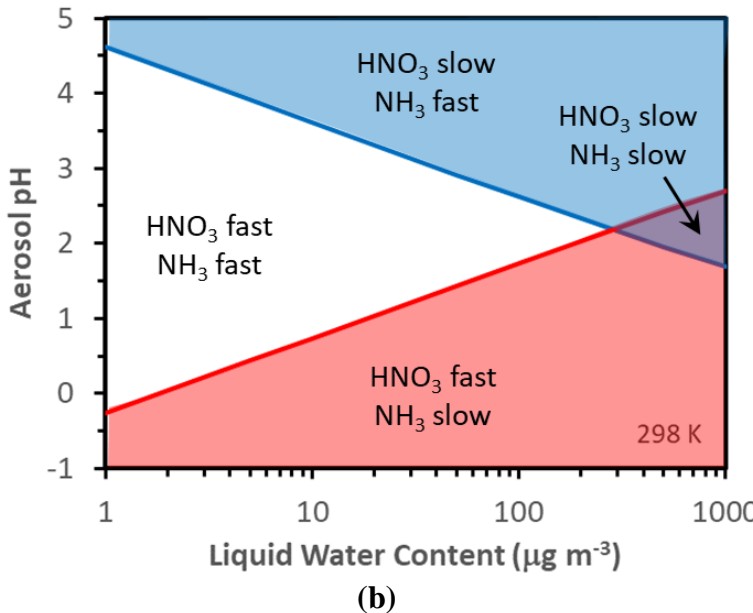

(b)

**Figure 2.**





Figure 3.


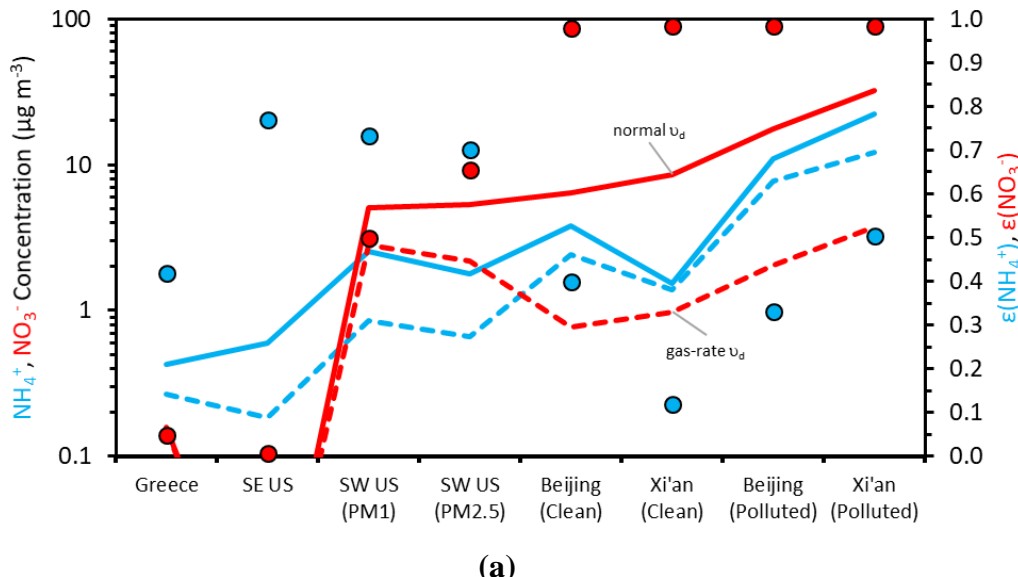

**(a)**

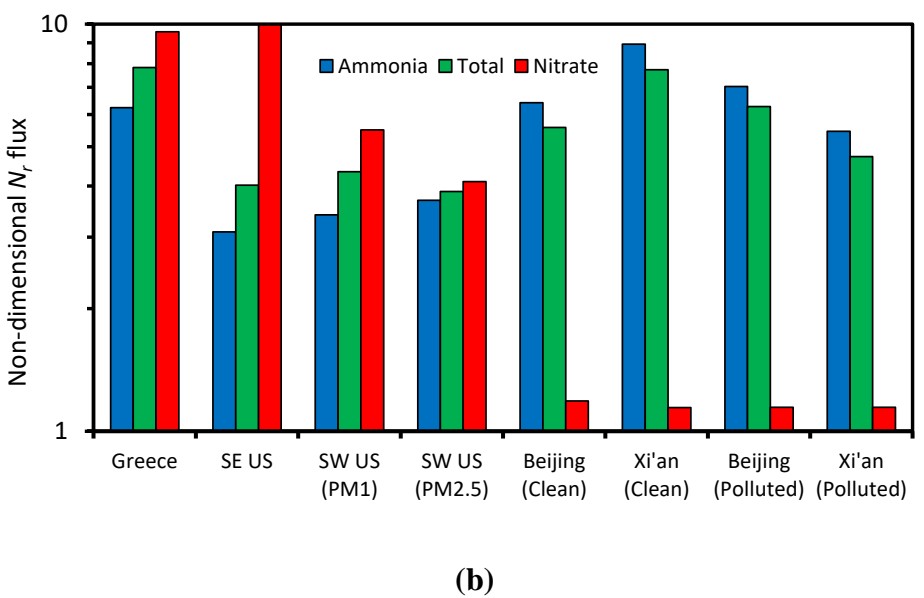

**(b)**

**Figure 4.**



**(a)**

**(b)**

**Figure 5.**