# Peer review of "Aerosol acidity and liquid water content regulate the dry deposition of inorganic reactive nitrogen"

_Atmospheric Chemistry and Physics, 2020_

## Referee Comment (RC1) · Anonymous Referee #1 · 3 Apr 2020

This is a nice study that develops an analytical framework for understanding how particle acidity and liquid water content indirectly modulate the rate of reactive nitrogen deposition through their influence on gas-particle partitioning of total nitrate and total ammonium. The work is presented in a very clear, simple way and makes use of valuable observational datasets from sites around the world to illustrate the application of the conceptual framework. Overall, the study improves how we think about the connections between aerosol thermodynamics and deposition and suggests how the framework might be used to better evaluate and characterize performance of the photochemical models widely used in decision making. The paper is a nice companion to the authors' 2020 ACP paper on how aerosol acidity and liquid water content determine

the sensitivity of particulate matter to the availability of ammonia and nitrate. I have a few comments below for the authors to consider.

General comments:

1. The authors define chemical regimes for deposition velocity that transition from "slow" to "fast" when the particle-phase fraction of the total amounts transition from greater than 90% to less than 90%. One might expect the regimes to vary from "slow" to "moderate" to "fast" as a function of the gas-particle partitioning and that fast deposition (associated with gas-phase deposition rates) would not occur until much more than 10% of the total is in the gas phase. It would be worthwhile for the authors to add a sentence or two describing the slow-fast transition and choice of alpha, beta=0.9. Is the idea that deposition will approach the gas-phase rate once appreciable amounts are in the gas phase because rapid gas-phase deposition would cause evaporation from particles to maintain equilibrium and ultimately rapid removal of the total?

2. The authors use a value of k=10 to represent the ratio of the deposition velocity of the gas phase to the deposition velocity of the particle phase. This is reasonable for the order-of-magnitude considerations presented, but ACP readers would likely be interested in some additional information and references on the deposition rates of gas and particle phase species. Can the authors provide some references for the values that led to the choice of k=10? It would also be interesting know ranges of values for typical variations in ambient conditions either from model predictions or the literature.

3. On p. 4, the authors mention their acidity/water framework for understanding the sensitivity of PM to "emissions of ammonia and nitrate". As the authors are aware, very little nitrate is directly emitted, and nitrate forms largely from oxidation of NOx emissions. A challenge in using the thermodynamic framework is that the conversion of NOx to HNO3 may be oxidant limited such that decreasing NOx emissions could potentially increase HNO3 concentrations. Such nonlinearities in gas-phase oxidation complicate the relationship between the thermodynamic framework and precursor

emissions. In the case of ammonia, changes in ammonia emissions influence cloud pH and therefore in-cloud production of sulfate, and the feedbacks of ammonia on this sulfate production are not accounted for in the framework. Given these complications, I recommend that the authors reword statements on p. 4 to avoid directly connecting the thermodynamic framework to the precursor emissions.

4. Is there any synergy in diurnal profiles of aerosol thermodynamics and deposition? For instance, partitioning to the particle phase tends to be greater during cooler conditions overnight with lower wind speeds, which would presumably influence aerodynamic resistance.

Specific comments:

– P. 1, line 27: clarify that NH3 and HNO3 are intended to represent "total" amounts, not just the gas phase

– P. 1, line 30: Sentence ends in comma, should be period

– P. 1, line 31: Should "reduce the deposition" be "reduce the local deposition" since all of the material will eventually deposit?

– P. 3, line 2: Should "NRC" be "NAS" to match the reference list?

– P. 3, line 11: Recommend changing "significant fraction of Nr" to "significant fraction of Nr deposition". NOx represents a large amount of oxidized nitrogen but a relatively smaller amount of Nr deposition because its deposition rate is relatively low.

– P. 4, line 1: recommend adding "for deposition" after "regimes"

– P. 4, lines 14-15: "velocity deposition" should be "deposition velocity"

– P. 4, Figure 1: can any intuition be provided on what 10 and 250 ug/m3 of water correspond to in terms of typical PM levels?

– P. 6, line 4: I think pH should be ">" than pH' in regime 3. Please check.

– P. 6, Figure 2: The larger "fast" region at higher temperature suggests that deposition would be slower and long-range transport more prevalent in winter. Might be worth mentioning the implications of the work for winter/summer differences somewhere.

– P. 7, eqn 5: v is missing the subscript "p"

– P. 8, Figure 4b: a linear (rather than log) scale seems more intuitive to me for the normalized flux plot. Please consider revising.

– P. 9, line 8: might want to indicate "total" in front of nitrate and ammonia for clarity

– P. 10, line 7: the nature of the "positive feedback" is not explicitly stated here and might not be clear to all

---

## Referee Comment (RC2) · Anonymous Referee #2 · 11 Jul 2020

This is an interesting and original manuscript based on scientifically sound investigation. The paper is well written and clearly structured. The paper should be published after the authors have considered the following minor issues.

Page 4, line 18: The authors state, without any justification, that the dry deposition both ammonia and nitric acid are about 10 times higher than the corresponding particulate forms of these compounds. The authors should shortly explain what this statement is based on? Dry deposition of aerosol particle is strongly dependent on particle size, so do the authors implicitly assume that both ammonium and nitrate are located in the typical accumulation mode size range? As mentioned briefly in section 6, a notable

fraction of nitrate can be in the coarse mode at high dust (or sea salt) loading, which would increase the deposition velocity of particulate nitrate.

When investigating and discussing the implication of this study, the authors emphasize air pollution accumulation. This is certainly an important implication. Another theme that the results of this study might have implications are the strength and spatial distribution of nitrogen (also acid) deposition. Could the authors elaborate this issue a bit more than just mentioning "prediction of nitrogen deposition flux" in one place (page 11, line 8)?

The caption of Figure 4a is insufficient to understand its contents, especially differences between the filled circles and lines. Figure 4a is first discussed in section 4, but it is not until section 5 where meaning of the lines in Figure 4a are shortly mentioned.

What are the 2 different columns under "Beijing" and "Xi'an" in Table 1? They are not explained in Table neither ei table caption. The same table has footnotes a and c not explained anywhere.

---

## Author Comment (AC1) · 30 Dec 2020

*Response to Reviewer #1 comments:*

This is a nice study that develops an analytical framework for understanding how particle acidity and liquid water content indirectly modulate the rate of reactive nitrogen deposition through their influence on gas-particle partitioning of total nitrate and total ammonium. The work is presented in a very clear, simple way and makes use of valuable observational datasets from sites around the world to illustrate the application of the conceptual framework. Overall, the study improves how we think about the connections between aerosol thermodynamics and deposition and suggests how the framework might be used to better evaluate and characterize performance of the photochemical models widely used in decision making. The paper is a nice companion to the authors' 2020 ACP paper on how aerosol acidity and liquid water content determine the sensitivity of particulate matter to the availability of ammonia and nitrate. I have a few comments below for the authors to consider.

*We thank the reviewer for the enthusiastic response and for feedback that has improved the manuscript. Below, we include the response to comments and questions raised.*

General comments:
1.  The authors define chemical regimes for deposition velocity that transition from "slow" to "fast" when the particle-phase fraction of the total amounts transition from greater than 90% to less than 90%. One might expect the regimes to vary from "slow" to "moderate" to "fast" as a function of the gas-particle partitioning and that fast deposition (associated with gas-phase deposition rates) would not occur until much more than 10% of the total is in the gas phase. It would be worthwhile for the authors to add a sentence or two describing the slow-fast transition and choice of alpha, beta=0.9. Is the idea that deposition will approach the gas-phase rate once appreciable amounts are in the gas phase because rapid gas-phase deposition would cause evaporation from particles to maintain equilibrium and ultimately rapid removal of the total?

    *Answer: We thank the reviewer for raising this point. Indeed, the deposition velocity changes between the particle and gas-phase limit over a range of ε. Although the feedback suggested does take place, the prime reason why we have chosen the thresholds is that in the map areas with slow-fast pairs (slow $NH_3$/Fast $HNO_3$ and vice-versa) deposition conditions most closely approach the gas-particle deposition velocity limits. This will be noted in the revised text.*

2.  The authors use a value of k=10 to represent the ratio of the deposition velocity of the gas phase to the deposition velocity of the particle phase. This is reasonable for the order-of-magnitude considerations presented, but ACP readers would likely be interested in some additional information and references on the deposition rates of gas and particle phase species. Can the authors provide some references for the values that led to the choice of k=10? It would also be interesting know ranges of values for typical variations in ambient conditions either from model predictions or the literature.

    *Answer: The reviewer raises a good point, and a similar comment was also raised by Reviewer 2. References will be noted as well as typical ranges in ambient conditions.*

3.  On p. 4, the authors mention their acidity/water framework for understanding the sensitivity of PM to "emissions of ammonia and nitrate". As the authors are aware, very little nitrate is

directly emitted, and nitrate forms largely from oxidation of NOx emissions. A challenge in using the thermodynamic framework is that the conversion of NOx to HNO3 may be oxidant limited such that decreasing NOx emissions could potentially increase HNO3 concentrations. Such nonlinearities in gas-phase oxidation complicate the relationship between the thermodynamic framework and precursor emissions. In the case of ammonia, changes in ammonia emissions influence cloud pH and therefore in-cloud production of sulfate, and the feedbacks of ammonia on this sulfate production are not accounted for in the framework. Given these complications, I recommend that the authors reword statements on p. 4 to avoid directly connecting the thermodynamic framework to the precursor emissions.

*Answer: We thank the reviewer for pointing out these valid concerns. We have reworded the relevant statements and added clarifications in the text to address the points raised.*

4. Is there any synergy in diurnal profiles of aerosol thermodynamics and deposition? For instance, partitioning to the particle phase tends to be greater during cooler conditions overnight with lower wind speeds, which would presumably influence aerodynamic resistance.

*Answer: There is certainly the aforementioned synergy. Nighttime conditions tend to be characterized by lower temperatures and higher humidity – both of which favor partitioning to the aerosol phase, which slows down deposition for both types of reduced nitrogen. This diurnal-deposition cycle exacerbates the daytime-nighttime contrast in boundary layer aerosol concentration – and will be included as an additional insight in the revised manuscript.*

**Specific comments:**
P. 1, line 27: clarify that NH3 and HNO3 are intended to represent "total" amounts, not just the gas phase

*Answer: This is now clear in the text.*

P. 1, line 30: Sentence ends in comma, should be period.

*Answer: Correction included.*

P. 1, line 31: Should "reduce the deposition" be "reduce the local deposition" since all of the material will eventually deposit?

*Answer: Indeed so! "Local" is now stated.*

P. 3, line 2: Should "NRC" be "NAS" to match the reference list?

*Answer: Done.*

P. 3, line 11: Recommend changing "significant fraction of Nr" to "significant fraction of Nr deposition". NOx represents a large amount of oxidized nitrogen but a relatively smaller amount of Nr deposition because its deposition rate is relatively low.

*Answer: Very good point. Clarification made as suggested.*

P. 4, line 1: recommend adding "for deposition" after "regimes"

*Answer: Done.*

P. 4, lines 14-15: "velocity deposition" should be "deposition velocity"

*Answer: Correction made.*

P. 4, Figure 1: can any intuition be provided on what 10 and 250 ug/m3 of water correspond to in terms of typical PM levels?

*Answer: Good point! We will provide characteristic levels of dry aerosol mass that would be associated with the water content noted.*

P. 6, line 4: I think pH should be ">" than pH' in regime 3. Please check.

*Answer: Indeed so. Thank you for noting this typo! Change made.*

P. 6, Figure 2: The larger "fast" region at higher temperature suggests that deposition would be slower and long-range transport more prevalent in winter. Might be worth mentioning the implications of the work for winter/summer differences somewhere.

*Answer: This is a great point; these insights will be noted.*

P. 7, eqn 5: v is missing the subscript "p"

*Answer: Thank you for pointing this out. Change made.*

P. 8, Figure 4b: a linear (rather than log) scale seems more intuitive to me for the normalized flux plot. Please consider revising.

*Answer: We have made the requested change.*

P. 9, line 8: might want to indicate "total" in front of nitrate and ammonia for clarity

*Answer: Indeed so. Changes made.*

P. 10, line 7: the nature of the "positive feedback" is not explicitly stated here and might not be clear to all

*Answer: We have now clarified this point in the revised text.*

---

## Author Comment (AC2) · 30 Dec 2020

*Response to Reviewer #2 comments:*

This is an interesting and original manuscript based on scientifically sound investigation. The paper is well written and clearly structured. The paper should be published after the authors have considered the following minor issues.

*We thank the reviewer for the enthusiastic response and constructive comments. Below, we include the response to comments and questions raised.*

General comments:
Page 4, line 18: The authors state, without any justification, that the dry deposition both ammonia and nitric acid are about 10 times higher than the corresponding particulate forms of these compounds. The authors should shortly explain what this statement is based on? Dry deposition of aerosol particle is strongly dependent on particle size, so do the authors implicitly assume that both ammonium and nitrate are located in the typical accumulation mode size range? As mentioned briefly in section 6, a notable fraction of nitrate can be in the coarse mode at high dust (or sea salt) loading, which would increase the deposition velocity of particulate nitrate.

> *Answer: The reviewer raises a good point. The deposition velocity indeed changes with size, but for all relevant sizes still differ between the gas and aerosol states up to particles of roughly 10 μm diameter, where it is roughly 1 cm s$^{-1}$ (e.g., Lin et al, 1994). The stated 10-fold velocity difference is indeed an average for submicron aerosol (e.g., Duyzer, JGR, 1994). The whole theory is based on the gas-to-particle velocity ratio k to account for any other factor that makes the deposition velocity differ. This discussion is now reflected in the text.*

When investigating and discussing the implication of this study, the authors emphasize air pollution accumulation. This is certainly an important implication. Another theme that the results of this study might have implications are the strength and spatial distribution of nitrogen (also acid) deposition. Could the authors elaborate this issue a bit more than just mentioning "prediction of nitrogen deposition flux" in one place (page 11, line 8)?

> *Answer: The concentration and spatial distribution of nitrogen species and their relation to acidity and liquid water changes is indeed central to this manuscript. The discussion about deposition patterns is mentioned in other places besides Page 11, as it is discussed in terms of long-range transport and expressed in terms of a conceptual model in the last figure of the paper. We will further elaborate on these points in the revised manuscript.*

The caption of Figure 4a is insufficient to understand its contents, especially differences between the filled circles and lines. Figure 4a is first discussed in section 4, but it is not until section 5 where meaning of the lines in Figure 4a are shortly mentioned.

> *Answer: This is a good point. We will further expand the caption to make it self-explanatory.*

What are the 2 different columns under "Beijing" and "Xi'an" in Table 1? They are not explained in Table neither ei table caption. The same table has footnotes a and c not explained anywhere.

> *Answer: We apologize for these oversights. "Beijing" and "Xi'an" refer to data from different locations in China. We will now explain this better in the table and also address the unresolved footnotes.*

**References**

Lin, J.J., Noll, K.E., Holsen, T.M. (1994) Dry Deposition Velocities as a Function of Particle Size in the Ambient Atmosphere, Aerosol Sci. Tech., 20:3, 239-252, doi:10.1080/02786829408959680

Duyzer, J. (1994) Dry deposition of ammonia and ammonium aerosols over heathland, J. Geophys. Res., 99, 18757-18763, D9, doi:10.1029/94JD01210